# PS-TTL: Prototype-based Soft-labels and Test-Time Learning for Few-shot Object Detection

## ABSTRACT

In recent years, Few-Shot Object Detection (FSOD) has gained widespread attention and made significant progress due to its ability to learn models with strong generalization power using extremely limited annotated data. Although the fine-tuning based paradigm for FSOD has become mainstream, where detectors are initially pretrained on base classes with sufficient samples and then fine-tuned on novel classes with few annotated samples, the scarcity of samples in novel classes hampers the precise capture of their data distribution. To address this issue, we propose a novel framework for FSOD, namely Prototype-based Soft-labels and Test-Time Learning (PS-TTL). Specifically, we design a Test-Time Learning (TTL) module that employs a mean-teacher network for self-training to discover novel instances on test data, effectively alleviating the problem of overfitting to the base class. Furthermore, we develop a Prototype-based Soft-labels (PS) strategy via assessing similarities between pseudo-labels and category prototypes to unleash the potential of low-quality pseudo-labels, thereby significantly mitigating the constraints posed by few-shot samples. Extensive experiments on both the VOC and COCO benchmarks show that PS-TTL achieves a new state-of-the-art, highlighting its effectiveness.

## CCS CONCEPTS

• **Computing methodologies** → **Scene understanding**; **Object detection**; **Online learning settings**.

## KEYWORDS

Few-shot Object Detection, Online Learning, Prototype, Pseudo Label

## 1 INTRODUCTION

Object detection [24, 39, 44, 50] is a fundamental computer vision task and has a variety of applications, including autonomous driving [58, 63], robotics [31, 37], medicine [23, 27], etc. Although significant progress has been archived in recent years [35, 43, 57, 60], these detectors heavily rely on a large number of training samples. On the other hand, humans can quickly extract novel concepts from a small amount of data. For example, children can learn to identify objects of novel categories after viewing a few pictures. The deep object detectors are also supposed to be able to learn effectively in data-limited scenarios because labelling data is quite expensive,

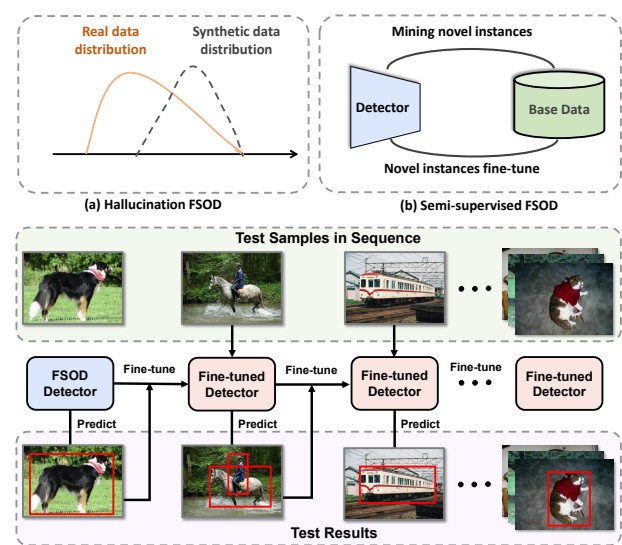

**Figure 1: Motivation of Test-Time Learning. (a) Hallucination FSOD methods suffer from mismatched distributions between synthetic data and real data. (b) Semi-supervised FSOD methods mine implicit novel instances from the base data; however, potential novel instances are not always included in the base data. (c) For the first time, rather than generating synthetic data for novel classes or mining implicit novel instances from the training set, we propose to learn at test time, effectively leveraging the novel class data present in the test data in a more realistic manner aligned with real-world applications.**

and collecting enough training examples for some rare categories is extremely hard.

Few-Shot Object Detection (FSOD) is a promising way to address this issue. It aims to train an object detector using only a few samples on novel classes with the help of abundant data on base classes, attracting widespread attention from researchers. Early FSOD methods typically adopt the meta-learning paradigm, organizing the task into a series of episodes simulating FSOD scenarios, where each episode includes few-shot training (support) and test (query) sets. The support set is utilized for model training with a limited number of samples, while the query set is employed to assess the model's detection performance on novel objects. Kang et al. [16] propose a lightweight feature reweighting module that learns to capture the global features of support images and embeds such features into reweighting coefficients to adjust the meta features of the query image. Meta R-CNN [52] develops a meta-learner, known as the Predictor-head Remodeling Network (PRN), leveraging a common Faster R-CNN [39] backbone to efficiently extract features from support images. Subsequently, meta-learning based

works progress from optimizing both classification and localization features [6, 9, 11, 46, 53], employing Transformer for capturing spatial relationships between support and query classes [13, 56], as well as exploring inter-class relationships [14, 17, 29, 59]. However, such methods involve complex architectures and training procedures, leading to increased computational complexity and costs. Additionally, they suffer from poor interpretability of what the model learned in the novel stage.

To facilitate faster training and simple deployment for rapid adaptation to novel classes, most existing FSOD methods employ a fine-tuning based paradigm. The detector is first pre-trained on base classes with adequate samples, then fine-tuned on novel classes with few annotated samples. Early methods [4, 48] employ a jointly fine-tuning based architecture, where the entire pre-trained base model, comprising both the class-agnostic and class-specific layers, is updated simultaneously during training on the novel task. Later, the two-stage fine-tuning approaches [2, 8, 32, 40, 45, 47, 62] demonstrate that maintaining the feature extraction part of the model unchanged and solely fine-tuning the last layer can significantly enhance detection accuracy. Based on this fact, most of the subsequent methods combined with knowledge distillation [33, 34, 49], context reasoning [19, 65], or decoupling detection networks [30, 36, 54] to further improve the detection performance. However, constrained by the limited samples for novel classes, they struggle to accurately capture the data distribution. Some works attempt to address this issue by generating synthetic data for novel classes [61, 64] or mining implicit novel instances from the training set [3, 18, 41]. However, the former method relies on information from base classes to synthesize novel samples, which may not accurately reflect the true distribution as shown in Fig. 1(a). The latter approach relies on the assumption that unlabeled novel instances are widely present in abundant base data as shown in Fig. 1(b), which may not hold true in real-world scenarios.

Considering the accessibility of novel instances in the test data, it motivates us to explore fine-tuning an object detection model at test-time as shown in Fig. 1(c). Compared to mining novel instances from base class data (the presence of unlabeled novel instances in base class data essentially represents a loophole in few-shot object detection settings), conducting online learning on test data is a more realistic approach aligned with real-world applications. In this paper, we first propose a Test-Time Learning (TTL) module, which utilizes a mean-teacher network for self-training to simultaneously train and test on test data, effectively leveraging the novel class data present in the test data. Specifically, both the student and teacher networks are first initialized by the FSOD detector fine-tuned on novel data. Then, the teacher network takes test data as input to generate pseudo-labels. The student model is trained using the pseudo-labels after post-processing and N-way-K-shot data as supervision signals and updates the teacher network through exponential moving average. Additionally, considering the limited number of high-quality pseudo labels and the fact that a large number of low-quality pseudo labels can recall most of the foreground but exhibit low classification accuracy, we develop a Prototype-based Soft-labels (PS) strategy to unlock the potential of these low-quality pseudo-labels. Specifically, we maintain class prototypes and compute the feature similarity between low-confidence pseudo-labels and class prototypes to replace

them with soft-labels. Class prototypes are initialized using N-way-K shot data and dynamically updated during online learning using the instance features of high-confidence pseudo-labels. Finally, we integrate the aforementioned two modules into a novel framework for few-shot object detection, dubbed PS-TTL.

In summary, the major contributions of this paper are:

- We propose a novel PS-TTL framework for few-shot object detection, which effectively mines new instances from test data to address the issue of limited novel class samples. To the best of our knowledge, it is the first attempt to explore fitting novel class data distributions in a way that is more in line with real-world scenarios.
- We design a Test-Time Learning (TTL) module that employs a mean-teacher network for self-training to discover novel instances on test data and develop a Prototype-based Soft-labels (PS) strategy to unleash the potential of low-quality pseudo-labels.
- We achieve a newly state-of-the-art performance of all few-shot settings on the VOC and COCO benchmarks in comparison to the published counterparts, and demonstrate its advantage in detecting novel objects.

## 2 RELATED WORK

### 2.1 Object Detection

Object detection aims to identify and localize objects within images, constituting a fundamental challenge in computer vision. Recently, the success of deep learning has yielded numerous effective object detection methods. These methods can be categorized into two main groups: two-stage and one-stage.

**Single-stage** detectors (e.g., SSD [26] and RetinaNet [24]) predict bounding boxes and classification scores based on predefined anchor boxes, exhibiting strong real-time performance. Subsequent anchor-free detectors [43, 60] alleviate the prior constraints of predefined anchors, further streamlining the detection process. The YOLO [38, 44] series, by continuously assimilating the latest advancements in object detection, such as label assignment and multi-scale feature fusion techniques, has achieved high-precision real-time object detection. Although the structure of single-stage detectors is straightforward, their integrated design also makes them less adaptable to FSOD tasks.

**Two-stage** detectors (e.g., Faster R-CNN [39] and Double-Head [50]) usually first use an region proposal network (RPN) to propose potential proposals, which are then refined by other modules. Methods such as Cascade R-CNN [1] and HTC [5] employ multi-stage refinement, further enhancing the detection precision. Compared to single-stage detectors, two-stage detection frameworks achieve higher detection performance. The concept of multi-stage refinement is also widely employed in the recently transformer-based detectors [35, 57]. Due to the design of multi-stage refinement, FSOD can achieve few-shot fine-tuning by controlling the gradients obtained by each stage of the detection module, effectively mitigating the issue of knowledge forgetting [45]. Two-stage detectors also facilitate the extraction of instance features for metric learning, making them commonly used in FSOD research [51].

## 2.2 Few-shot Object Detection

The FSOD methods enable detectors to swiftly adapt to new tasks with minimal data while preserving their original performance, enhancing the adaptability of models under data-constrained circumstances. FSOD methods can be broadly categorized into two paradigms: meta-learning based and fine-tuning based.

**Meta-learning based** methods employ numerous N-way K-shot detection tasks [55] for training, aiming to quickly adapt to new tasks with few support samples. FSRW [16] and Meta R-CNN [52] propose feature reweighting strategies on single-stage and two-stage detectors, respectively. They extract class-specific representations from support images and combine them with weighted queries to achieve detection for specific categories. Attention-RPN [9] integrates support information into the Region Proposal Network (RPN) and employs a contrastive training strategy to enhance the relevance between proposals and support classes. MetaDet [46] disentangle the learning of category-agnostic and category-specific components in detectors better to tackle few-shot classification and localization in a unified way. Recent efforts to improve meta-learning based approaches include introducing metric learning to enhance feature discriminability. CME [22] utilizes class margin loss to preserve sufficient margin space for novel classes. TIP [21] introduces consistency regularization on image transformation to enhance the model's generalization ability. Meta-learning methods enable detectors to rapidly adapt to new categories. However, the training and inference processes of these methods are highly complex, making them challenging to deploy in real-world scenarios.

**Fine-tuning based** methods adopt the two-stage training strategy, i.e., base training and then few-shot fine-tuning, which expects to transfer the prior knowledge from base classes to the novel classes. LSTD [4] is the earliest method to employ the two-stage training strategy for FSOD, using regularization to retain base knowledge. TFA [45] simply freezes the backbone and only fine-tunes the box classifier with instance-level normalization. Based on TFA, FSCE [40] introduces contrastive learning to learn the discriminative object proposal representations, alleviating the misclassification issues in novel classes. Subsequent research refine the TFA method and integrate it with other techniques to further enhancing the FSOD performance [10, 19, 30, 33, 34, 36, 49, 54, 65]. DeFRCN [36] employs the gradient decoupled layer to stop the gradient from the detector head, aiming to preserve generic knowledge of base classes while gradually extracting novel information in examples. PTF [54] devise an effective method for initializing novel class weights and propose an adaptive length re-scaling strategy to enhance classification precision. Although fine-tuning-based methods are simple for deployment, their generalization primarily relies on extensive pretraining with base data. When the disparity between novel and base classes is significant, detectors still struggle to adapt to novel domains with few samples.

**Semi-supervised learning** has been applied in FSOD to enrich the diversity of novel samples and address the issue of inconsistent label assignment for novel classes [3, 18, 41]. Kaul et al. [18] introduce a simple pseudo labelling strategy to detect potential novel instances in the base dataset. MINI [3] introduces a teacher-student framework and performs online parameter updates, enabling better novel instance mining. Tang et al. [41] propose a class-adaptive threshold filtering strategy to select more valuable pseudo labels. The core assumption of these methods is that novel instances appear frequently in the base dataset, which may not hold in real-world scenarios. However, novel instances are guaranteed to appear in the test set, making our TTL method more practically valuable.

## 3 METHODS

In this section, we initially review the problem setting of conventional few-shot object detection in Section 3.1, followed by a brief introduction to our baseline method, DeFRCN [36], in Section 3.2. Subsequently, we elaborate on our Test-Time Learning (TTL) module in Section 3.3 and Prototype-based Soft-labels (PS) strategy in Section 3.4. Finally, we delve into the training process of the entire framework in Section 3.5.

## 3.1 Problem Setting

We follow the standard few-shot object detection setting introduced in [45]. There are two disjoint training sets: a base dataset $D^b = \{x_i^b, y_i^b\}$ with exhaustively annotated instances for each base class $C^b$ and a novel dataset $D^n = \{x_i^n, y_i^n\}$ with only $K$ (usually less than 30) instances for each novel class $C^n$. In which $x_i$ and $y_i$ refer to the input image and the ground truth, respectively. It is worth noting that there is no intersection between the base classes and the novel classes, i.e., $C_b \cap C_n = \emptyset$. Therefore, the ultimate goal of the FSOD is to train a robust detector based on the $D^b$ and $D^n$ to detect objects in the test set that contains both instances in $C_b \cup C_n$.

## 3.2 Review of Few-shot Object Detector

DeFRCN [36] is a state-of-the-art fine-tuning based few-shot object detector, consisting of two training stages. In the first phase, the Faster-RCNN is trained on the base classes $C^b$ with sufficient samples. In the second phase, the transfer learning is performed, by fine-tuning the Faster-RCNN on the base classes and novel classes $C^b \cup C^n$ with $K$ instances per class. Fine-tuning on a balanced set $D^{nk}$ containing training samples for base and novel classes can help preserve the performance on base classes. The overall procedure of the fine-tuning based methods is summarized as follows:

$$F_{init} \xrightarrow{D^b} F_{base} \xrightarrow{D^{nk}} F_{novel} \qquad (1)$$

where $F_{init}$, $F_{base}$, and $F_{novel}$ indicate the detector in the initialization, base training, and novel fine-tuning stages, respectively.

Different from previous fine-tuning based methods, which only fine-tune a small number of parameters of the Faster-RCNN, such as the prediction head, to prevent overfitting of the detector. DeFRCN introduces a Gradient Decoupled Layer during fine-tuning to stop the gradient between RPN and backbone while scaling the gradient between RCNN and backbone. It allows the detector to learn sufficiently about the novel data while preventing overfitting and is remarkably superior to other existing approaches.

Despite the significant progress made by the fine-tuning based methods, given only $K$ novel instances, researchers fail to capture the data distribution accurately. To overcome the obstacles, we propose Prototype-based Soft-labels and Test-Time Learning (PS-TTL) to mine novel instances in the test data. The overall architecture of the model is illustrated in Fig. 2.

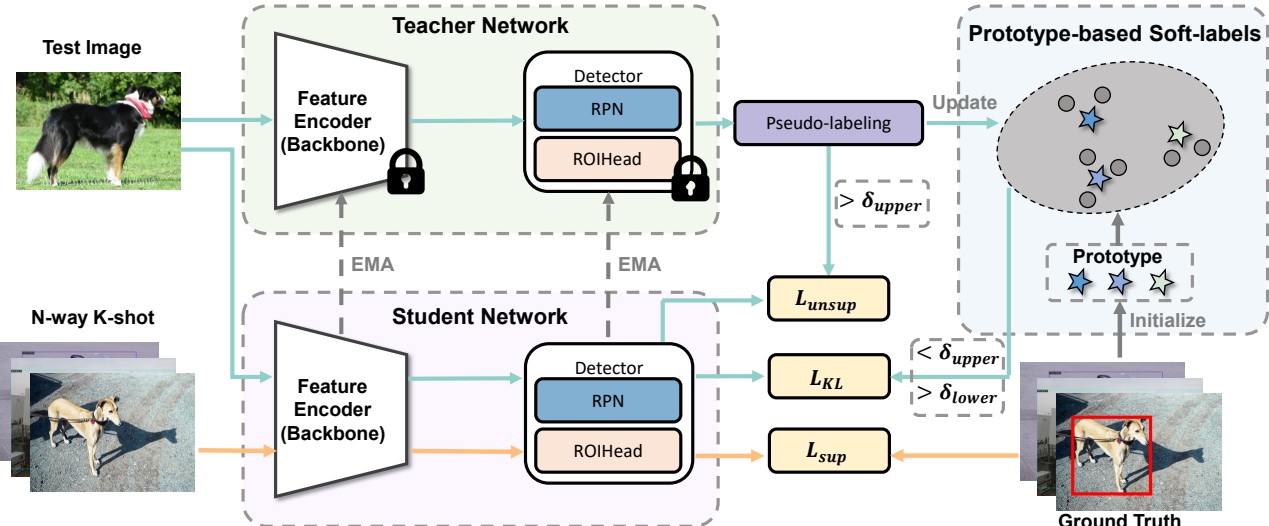

**Figure 2: The overview of the proposed Prototype-based Soft-labels and Test-Time Learning (PS-TTL) framework for FSOD. Both the student and teacher networks are first initialized by the FSOD detector and fine-tuned on novel data. Then, the teacher network takes test data as input to generate pseudo-labels. The student model is trained using the pseudo-labels after post-processing and N-way-K-shot data as supervision signals and updates the teacher network through exponential moving average. Additionally, a Prototype-based Soft-labels (PS) strategy is adopted to maintain class prototypes and compute the feature similarity between low-confidence pseudo-labels and class prototypes to replace them with soft-labels.**

## 3.3 Test-Time Learning with Mean-Teacher

Self-training has promising performance for semi-supervised object detection [20, 28, 42]. It typically generates pseudo labels for the unlabeled data, and then the high-confidence pseudo labels are used to supervise the detector training.

In this work, we hope to fully leverage novel instances in the test data, especially in the scenario of online learning, called Test-Time Learning (TTL). Therefore, we employ a mean-teacher self-training paradigm [42]. This paradigm mainly consists of two architecturally identical detectors, namely the student network and the teacher network. The teacher network first detects objects in the test data. Then we can obtain pseudo labels from the detection results through some post-processing procedures (e.g., non-maximum suppression and filtering using a confidence threshold). The high-quality pseudo labels are used to supervise the student network, enhancing its detection capability.

Since we rely on the teacher network to predict reliable pseudo labels of novel classes in the test data, we utilize the FSOD detector $F_{novel}$ which has fine-tuned on novel data as the initialization of both the student and teacher network. However, the self-training paradigm inevitably generates noisy pseudo labels, especially in the novel classes. If we use excessively noisy pseudo labels to train the student network, the performance of the detector would deteriorate as training progresses. To filter out the noisy pseudo labels, we first apply non-maximum suppression for each class to remove duplicate detection boxes. Then, we set a high confidence threshold $\delta_{upper}$ to exclude uncertain labels. Finally, we optimize the student network using the remaining high-quality pseudo labels with the loss function as follows:

$$L_{unsup}(X_t, \hat{Y}_t) = L_{cls}^{rpn}(X_t, \hat{Y}_t) + L_{cls}^{roi}(X_t, \hat{Y}_t) \qquad (2)$$

where $X_t$ is the input test image, and $\hat{Y}_t$ denotes the filtered pseudo labels. Note that the unsupervised loss is only applied to the classification heads of the Region Proposal Network (RPN) and Region of Interest (ROI) head.

Even after filtering out low-confidence predictions, the pseudo labels are still noisy due to the poor detection performance of the FSOD detector $F_{novel}$. Therefore, to alleviate the degradation of the FSOD detector during test-time learning, we propose using N-way-K-shot data $D^{nk}$ as supervision signals. Hence, the supervised loss for training the student network can be defined as:

$$L_{sup}(X_s, Y_s) = L_{cls}^{rpn}(X_s, Y_s) + L_{reg}^{rpn}(X_s, Y_s)$$
$$+ L_{cls}^{roi}(X_s, Y_s) + L_{reg}^{roi}(X_s, Y_s) \qquad (3)$$

Where $\{X_s, Y_s\} \in D^{nk}$. Both RPN and ROI head adopt classification loss and bounding box regression loss.

Following the mean-teacher [42], to obtain strong pseudo labels from the test data, we update the teacher network weights via Exponential Moving Average (EMA) of student ones as below:

$$\theta_t = \alpha\theta_t + (1 - \alpha)\theta_s \qquad (4)$$

where $\theta_t$ and $\theta_s$ are the network parameters of the teacher network and the student network, respectively. And $\alpha$ is the EMA momentum coefficient.

## 3.4 Prototype-based Soft-labels Strategy

Utilizing the mean-teacher self-training framework proposed in Section 3.3 for test-time learning on the test data can promote the detection performance. Through experiments, we found that it is necessary to choose a large threshold $\delta_{upper}$ to filter the generated pseudo labels. However, this leads to severely missed detections,

indicating that many test images have no pseudo labels. Different from semi-supervised object detection, where multiple rounds of fine-tuning can be conducted on the unlabeled data. Under the test-time learning setting, we can only perform one epoch of training on the test data. How to fully utilize every input test image is crucial.

As shown in the Fig. 3, we observed that relatively low-confidence pseudo labels, despite having classification confusion, mostly recall the foreground. Based on this phenomenon, we propose a Prototype-based Soft-labels (PS) strategy to replace the hard labels of these implicit foreground predictions with soft labels for fully unleashing the potential of low-quality pseudo-labels.

Firstly, we introduce a lower bound confidence threshold $\delta_{lower}$; the predicted results between $\delta_{lower}$ and $\delta_{upper}$ are also assigned as foreground. Due to the increased class confusion in these implicit foreground predictions, employing class-specific NMS in the teacher network fails to effectively remove redundant boxes. Therefore, after removing the hard labels of these implicit foreground predictions, we apply class-agnostic NMS to them using every high-confidence pseudo prediction (i.e., whose confidence score is greater than $\delta_{upper}$) to filter the redundant ones.

We then generate soft labels for the implicit foreground predictions by measuring their similarities to each class. Formally, given a implicit foreground prediction $r$, we define its similarity to a class $c$ as the cosine distance between its ROI feature $V_r$ and the prototype $P_c$ of the class $c$:

$$s_r^c = \frac{V_r^T P_c}{||V_r^T|| \, ||P_c||}, \quad c \in C^b \cup C^n \tag{5}$$

Finally, $s_r = [s_r^1, s_r^2, ..., s_r^N]$ followed a softmax function to generate $q_r$, which represents the soft label of the implicit foreground prediction $r$. And we minimize the Kullback-Leibler (KL) divergence between the soft label and the class logits $p_r$ of each implicit foreground prediction $r$:

$$L_{KL} = \sum_{c=1}^{N+1} q_r^c log(\frac{q_r^c}{p_r^c}) \tag{6}$$

where $N+1$ denotes $N$ foreground classes and one background class. Additionally, we set $q_r^{N+1} = 0$.

To leverage soft labels of the implicit foreground predictions at the early stage during test-time learning, we initialize the class prototypes with N-way-K-shot data:

$$P_c = \frac{1}{K} \sum_{i=1}^{K} f_c^i \tag{7}$$

where $f_c^i$ is the ROI feature of the $i$-th instance for class $c$. Because N-way-K-shot data cannot accurately represent the class prototypes, we propose dynamically updating the class prototypes using both labeled data and test data with high-confidence pseudo labels, aiming for the class prototypes to converge to the true representations as training progresses. Specifically, we update the class prototypes using the following formula:

$$P_c = P_c(1 - sim(P_c, \overline{f_c})) + \overline{f_c} sim(P_c, \overline{f_c}) \tag{8}$$

where $\overline{f_c}$ is the averaged ROI features for class $c$. And $sim(\cdot, \cdot)$ is the cosine similarity function.

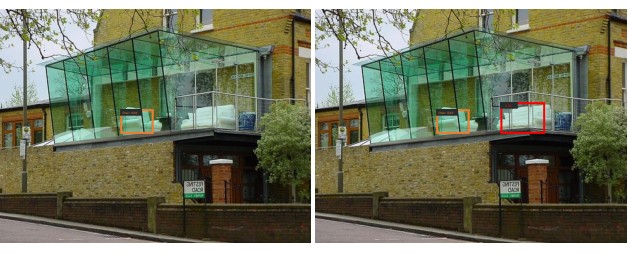

**Figure 3: Illustration for the issue of low-confidence pseudo labels. In the left image, pseudo-labels are generated using $\delta_{upper}$ filtering for self-training. In the right image, as we decrease the threshold, the low-confidence pseudo labels are converted into high-quality implicit foreground predictions.**

## 3.5 Training Procedure

During test-time learning, the total loss we optimize is as follows:

$$L_{total} = L_{sup} + \lambda_1 L_{unsup} + \lambda_2 L_{KL} \tag{9}$$

which consists of supervised loss of N-way-K-shot data, unsupervised loss of pseudo labels, and KL loss of soft labels. Here, $\lambda_1$ and $\lambda_2$ are hyper-parameters to balance among losses.

Then, we summarize the overall algorithm. We aim for the FSOD detector to learn from the test data. When a mini-batch of testing samples arrives, we update the model weights through the total loss. A detailed description is provided in Algorithm 1.

---

**Algorithm 1:** PS-TTL algorithm.

---

**Input:** Testing data $\mathcal{D}^t = \{x_i^t\}_{i=1 \cdots N_t}$, and N-way K-shot data $\mathcal{D}^{nk} = \{x_i, y_i\}_{i=1 \cdots nk}$

**for** $x_i^t \leftarrow 1$ **to** $N_t$ **do**

  # Inference Stage:

  Predict objects: $Res = $ teacherDet$(x_i^t)$

  # Fine-tune Stage:

  Get supervised data: $\{x_i, y_i\} = $ iter$(\mathcal{D}^{nk}, i)$

  Pseudo label prediction: $\mathcal{P} = $ teacherDet$(x_i^t)$;

  Pseudo label filter: threshold$(\mathcal{P})$&NMS$(\mathcal{P})$

  Total loss: $\mathcal{L}_{total} = \mathcal{L}_{sup} + \lambda_1 \mathcal{L}_{unsup} + \lambda_2 \mathcal{L}_{KL}$;

  Student update (Gradient descent):

  $\Theta_s = \Theta_s - \eta \nabla \mathcal{L}_{total}$

  Teacher update (EMA):

  $\Theta_t = \alpha \Theta_t + (1 - \alpha)\Theta_s$

---

## 4 EXPERIMENTS

In this section, we first introduce the experimental benchmarks in Section 4.1 and then describe the implementation details of our method in Section 4.2. Following this, we conduct extensive experiments on PASCAL VOC and MS COCO to compare our method with previous state-of-the-art approaches in Section 4.3. Finally, we provide extensive ablation studies on different components in Section 4.4.

**Table 1: Comparison of different few-shot object detection methods in terms of nAP50 on three PASCAL VOC Novel Split sets.**

| Method / Shots | Novel Split 1 | | | | | Novel Split 2 | | | | | Novel Split 3 | | | | |
|---|---|---|---|---|---|---|---|---|---|---|---|---|---|---|---|
| | 1 | 2 | 3 | 5 | 10 | 1 | 2 | 3 | 5 | 10 | 1 | 2 | 3 | 5 | 10 |
| YOLO-ft [16] | 6.6 | 10.7 | 12.5 | 24.8 | 38.6 | 12.5 | 4.2 | 11.6 | 16.1 | 33.9 | 13.0 | 15.9 | 15.0 | 32.2 | 38.4 |
| FRCN-ft [52] | 13.8 | 19.6 | 32.8 | 41.5 | 45.6 | 7.9 | 15.3 | 26.2 | 31.6 | 39.1 | 9.8 | 11.3 | 19.1 | 35.0 | 45.1 |
| LSTD [4] | 8.2 | 1.0 | 12.4 | 29.1 | 38.5 | 11.4 | 3.8 | 5.0 | 15.7 | 31.0 | 12.6 | 8.5 | 15.0 | 27.3 | 36.3 |
| FSRW [16] | 14.8 | 15.5 | 26.7 | 33.9 | 47.2 | 15.7 | 15.3 | 22.7 | 30.1 | 40.5 | 21.3 | 25.6 | 28.4 | 42.8 | 45.9 |
| MetaDet [46] | 18.9 | 20.6 | 30.2 | 36.8 | 49.6 | 21.8 | 23.1 | 27.8 | 31.7 | 43.0 | 20.6 | 23.9 | 29.4 | 43.9 | 44.1 |
| Meta R-CNN [52] | 19.9 | 25.5 | 35.0 | 45.7 | 51.5 | 10.4 | 19.4 | 29.6 | 34.8 | 45.4 | 14.3 | 18.2 | 27.5 | 41.2 | 48.1 |
| RepMet [17] | 26.1 | 32.9 | 34.4 | 38.6 | 41.3 | 17.2 | 22.1 | 23.4 | 28.3 | 35.8 | 27.5 | 31.1 | 31.5 | 34.4 | 37.2 |
| TFA w/cos [45] | 39.8 | 36.1 | 44.7 | 55.7 | 56.0 | 23.5 | 26.9 | 34.1 | 35.1 | 39.1 | 30.8 | 34.8 | 42.8 | 49.5 | 49.8 |
| MPSR [48] | 41.7 | – | 51.4 | 55.2 | 61.8 | 24.4 | – | 39.2 | 39.9 | 47.8 | 35.6 | – | 42.3 | 48.0 | 49.7 |
| HallucFsDet [61] | 47.0 | 44.9 | 46.5 | 54.7 | 54.7 | 26.3 | 31.8 | 37.4 | 37.4 | 41.2 | 40.4 | 42.1 | 43.3 | 51.4 | 49.6 |
| Retentive R-CNN[10] | 42.4 | 45.8 | 45.9 | 53.7 | 56.1 | 21.7 | 27.8 | 35.2 | 37.0 | 40.3 | 30.2 | 37.6 | 43.0 | 49.7 | 50.1 |
| FSCE [40] | 44.2 | 43.8 | 51.4 | 61.9 | 63.4 | 27.3 | 29.5 | 43.5 | 44.2 | 50.2 | 37.2 | 41.9 | 47.5 | 54.6 | 58.5 |
| SRR-FSD [65] | 47.8 | 50.5 | 51.3 | 55.2 | 56.8 | 32.5 | 35.3 | 39.1 | 40.8 | 43.8 | 40.1 | 41.5 | 44.3 | 46.9 | 46.4 |
| CME [22] | 41.5 | 47.5 | 50.4 | 58.2 | 60.9 | 27.2 | 30.2 | 41.4 | 42.5 | 46.8 | 34.3 | 39.6 | 45.1 | 48.3 | 51.5 |
| FADI [2] | 50.3 | 54.8 | 54.2 | 59.3 | 63.2 | 30.6 | 35.0 | 40.3 | 42.8 | 48.0 | 45.7 | 49.7 | 49.1 | 55.0 | 59.6 |
| UP-FSOD [47] | 43.8 | 47.8 | 50.3 | 55.4 | 61.7 | 31.2 | 30.5 | 41.2 | 42.2 | 48.3 | 35.5 | 39.7 | 43.9 | 50.6 | 53.3 |
| QA-FewDet [11] | 42.4 | 51.9 | 55.7 | 62.6 | 63.4 | 25.9 | 37.8 | 46.6 | 48.9 | 51.1 | 35.2 | 42.9 | 47.8 | 54.8 | 53.5 |
| LVC‡ [18] | 54.5 | 53.2 | 58.8 | 63.2 | 65.7 | 32.8 | 29.2 | 50.7 | 49.8 | 50.6 | 48.4 | 52.7 | 55.0 | 59.6 | 59.6 |
| DeFRCN* [36] | 55.4 | 62.1 | 65.0 | 68.4 | 67.6 | 35.5 | 45.4 | 51.8 | 51.7 | 47.5 | 50.8 | 57.4 | 57.8 | 62.7 | **65.0** |
| Ours | **58.4** | **65.5** | **67.9** | **69.3** | **68.1** | **38.4** | **47.8** | **52.8** | **53.6** | **49.1** | **53.0** | **58.8** | **59.2** | **63.8** | 64.1 |

**Table 2: Few-shot object detection performance on MS COCO.**

| Method | 10-shot | | 30-shot | |
|---|---|---|---|---|
| | nAP | nAP75 | nAP | nAP75 |
| FSRW [16] | 5.6 | 4.6 | 9.1 | 7.6 |
| MetaDet [46] | 7.1 | 6.1 | 11.3 | 8.1 |
| Meta R-CNN [52] | 8.7 | 6.6 | 12.4 | 10.8 |
| DCNet [15] | 12.8 | 11.2 | 18.6 | 17.5 |
| CME [22] | 15.1 | 16.4 | 16.9 | 17.8 |
| TFA [45] | 9.1 | 8.8 | 12.1 | 12.0 |
| MPSR [48] | 9.8 | 9.7 | 14.1 | 14.2 |
| Retentive R-CNN [10] | 10.5 | – | 13.8 | – |
| FSCE [40] | 11.4 | 10.1 | 15.8 | 14.7 |
| SRR-FSD [65] | 11.3 | 9.8 | 14.7 | 13.5 |
| FADI [2] | 12.2 | – | 16.1 | – |
| QA-FewDet [11] | 11.6 | 9.8 | 16.5 | 15.5 |
| Meta FRCN [12] | 9.7 | 9.0 | 10.7 | 10.6 |
| VFA [14] | 16.2 | – | 18.9 | – |
| LVC‡ [18] | 18.6 | 18.5 | 26.1 | 26.8 |
| DeFRCN* [36] | 17.1 | 15.9 | 20.2 | 19.5 |
| Ours | **17.3** | **16.7** | **20.9** | **21.3** |

## 4.1 Datasets

**PASCAL VOC.** For PASCAL VOC [7], the overall 20 classes are divided into 15 base classes and 5 novel classes. Following TFA [45], we utilize three different class splits, namely split 1, 2, and 3.

For each split, base classes are exhaustively annotated, but novel classes only have $K = 1, 2, 3, 5, 10$ annotated instances per class. Both base and novel class instances are sampled from the PASCAL VOC (07+12) trainval set, and the model is tested on the PASCAL VOC07 test set. We report AP50 for novel classes during evaluation.

**MS COCO.** MS COCO [25] has 80 classes, we selecte the 20 classes that overlapped with PASCAL VOC as novel classes and the remaining 60 classes as base classes. In this case, we evaluate our method with $K = 10, 30$ shots for each novel class. And we report mAP, and AP75, respectively.

## 4.2 Implementation Details

Our method can be combined with majority fine-tuning based few-shot object detector. For simplicity, we chose the most representative SOTA method, DeFRCN [36], as our baseline. DeFRCN uses Faster-RCNN [39] as the detection model and ImageNet pre-trained ResNet-101. We use DeFRCN, which has been pre-trained on base classes and fine-tuned on novel classes, as the initialization of our model, and then fine-tune on the test data. During test-time learning, we fine-tune our model with a mini-batch of 2 on single GPU, which simulate the real inference process of the FSOD detector. Besides, we adopt a one-pass setting, where we fine-tune on the test data for only one epoch. We also utilize the N-way K-shot data used for novel fine-tuning during the testing process. Due to the poor performance of the FSOD detector, we apply weak data augmentation to both the N-way K-shot data and the test data, including random resize and random horizontal flip. For the hyperparameter, we set the $\lambda_1 = 0.5$ and $\lambda_2 = 0.1$ for all the experiments for simplicity. We set the thresholds $\delta_{upper} = 0.9$ and $\delta_{lower} = 0.7$. We optimize the network

**Table 3: Contributions of each component to PS-TTL.**

| $L_{sup}$ | $L_{unsup}$ | $L_{KL}$ | nAP50 | | |
|:---:|:---:|:---:|:---:|:---:|:---:|
| | | | 1-shot | 2-shot | 3-shot |
| | | | 55.4 | 62.1 | 65.0 |
| ✓ | | | 54.3 | 61.5 | 63.2 |
| | ✓ | | 56.1 | 63.4 | 65.7 |
| ✓ | ✓ | | 57.0 | 63.8 | 65.4 |
| ✓ | ✓ | ✓ | **58.4** | **65.5** | **67.9** |

**Table 4: Ablation study of the threshold selection.**

| $\delta_{upper}$ | $\delta_{lower}$ | nAP50 | | |
|:---:|:---:|:---:|:---:|:---:|
| | | 1-shot | 2-shot | 3-shot |
| – | – | 55.4 | 62.1 | 65.0 |
| 0.95 | – | 55.5 | 61.6 | 64.2 |
| 0.90 | – | **57.0** | **63.8** | 65.4 |
| 0.85 | – | 56.8 | 63.4 | **65.6** |
| 0.90 | 0.8 | 57.1 | 65.2 | 67.4 |
| 0.90 | 0.7 | **58.4** | **65.7** | **67.9** |
| 0.90 | 0.6 | 57.5 | 65.2 | 67.5 |

using Stochastic Gradient Descent (SGD) and set the learning rate to 0.00125. The momentum coefficient of the EMA for the teacher network is set to 0.9996.

## 4.3 Main Results

**PASCAL VOC.** Experimental results on the PASCAL VOC dataset are shown in Table 1. We use DeFRCN as our baseline, which incorporates an additional Prototypical Calibration Block (PCB) for refining the predictions. However, we find that the N-way K-shot data utilized by the PCB may not align with that used during the novel fine-tuning stage. Therefore, we exclude the PCB and present our re-implementation results DeFRCN* in Tables 1 and 2. It can be observed that our method achieves improvements across various splits and different shots on PASCAL VOC benchmark. Our method outperforms HallucFsDet[61] and LVC[18], which represent synthetic novel class data and semi-supervised learning on the base data, respectively. Meanwhile, we find that the improvement gained from test-time learning becomes more significant as the shot decreases, especially in the 1-shot scenario.

**MS COCO.** Table 2 shows the detection results on MS COCO. The MS COCO dataset contains more categories, and typically, a single image contains multiple instances. FSOD detectors generally perform poorly on MS COCO due to these factors, which undermine the performance of our method. However, we observe that our method achieves a significant improvement compared to the baseline, especially in the mAP75 metric. There is a 5.0% improvement in AP75 at 10 shots and a 9.2% improvement in AP75 at 30 shots. LVC [18] demonstrates a noticeable improvement on the MS COCO dataset, because the base data in the MS COCO benchmark contains a large number of implicit novel instances. However, this issue arises from the setting of few-shot detection, which could not reflect the

**Table 5: Ablation study of the class prototypes update.**

| Update Methods | nAP50 | | |
|:---:|:---:|:---:|:---:|
| | 1-shot | 2-shot | 3-shot |
| Static | 57.5 | 65.1 | 67.7 |
| Dynamic | **58.4** | **65.7** | **67.9** |

**Table 6: Ablation study of the different data augmentation.**

| Student Aug. | Teacher Aug. | nAP50 | | |
|:---:|:---:|:---:|:---:|:---:|
| | | 1-shot | 2-shot | 3-shot |
| Strong | Weak | 56.9 | 64.9 | 66.6 |
| Weak | Weak | **58.4** | **65.7** | **67.9** |

real-world scenario. Under the test-time learning setting, we only have 5000 images available for mining the implicit novel instances.

## 4.4 Ablation Studies

In this section, we conduct ablation studies on novel split 1 of the PASCAL VOC benchmark to reveal the effectiveness of each individual component.

*4.4.1 Effectiveness of each component.* We conduct a detailed ablation study on each component of the model, as shown in Table 3. The first row presents our baseline, which is the result of DeFRCN. Initially, we attempted to solely utilize N-way K-shot data for supervised learning during testing but found that the model tended to overfit to these K-shot data, resulting in decreased performance. In the third row, we only fine-tuned the model using high-quality pseudo labels during testing, yielding results superior to the baseline. To further enhance FSOD performance in low-sample scenarios, we combined N-way K-shot data with pseudo-labels for training. Interestingly, except for the 3-shot setting, the model achieved further optimization in other cases, suggesting that this training approach effectively prevented the accumulation of biases in the model. Finally, by introducing $L_{KL}$, i.e., employing a prototype-based soft-label strategy during testing, the model significantly improved its performance across various sample sizes. This also indicates that our proposed method can more efficiently utilize pseudo-labels.

*4.4.2 Upper and lower thresholds setting.* Threshold selection has always been crucial in pseudo-label training, so we conducted ablation experiments on pseudo-label thresholds, as shown in Table 4. Firstly, we used a large threshold $\delta_{upper}$ to filter high-quality pseudo-labels as hard labels for training the student network. To determine the appropriate value of $\delta_{upper}$, we performed standard self-training on the test data without using soft labels. From Table 4, it can be observed that using a larger threshold may result in only a few pseudo-labels available as hard labels, which could lead to many foreground objects being mistakenly classified as background, damaging the model's detection performance. Conversely, setting a threshold too low introduces excessive noise labels, which also affects performance. By comparing the results from rows 1 to 4, we set $\delta_{upper}$ to 0.9. Next, we conducted experiments on the low threshold $\delta_{lower}$, where prediction boxes with confidence scores between the high threshold $\delta_{upper}$ and the low threshold $\delta_{lower}$ were

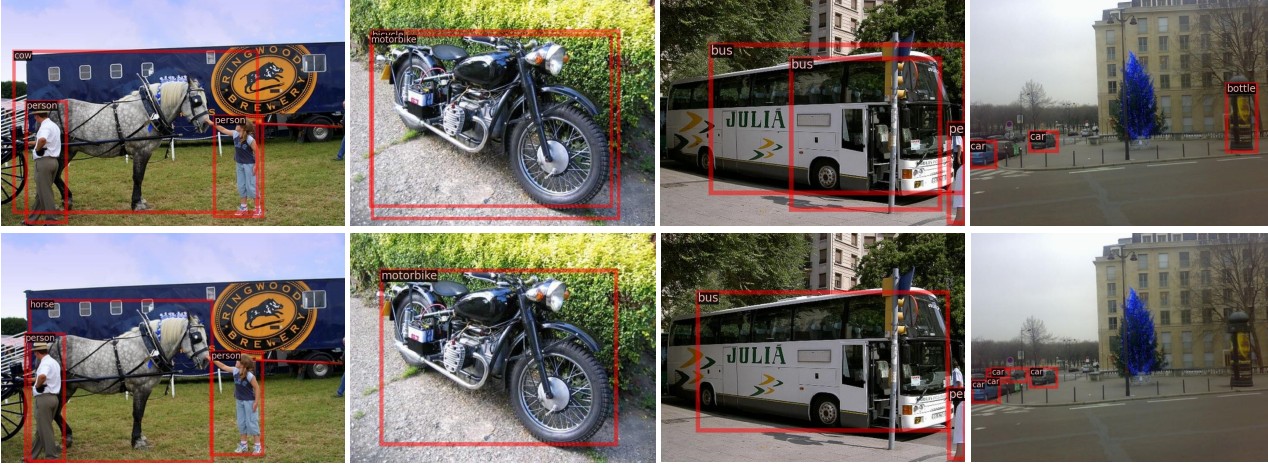

**Figure 4: Qualitative visualization comparison on PASCAL VOC. The top and bottom lines respectively show the detection results from DeFRCN and our PS-TTL.**

considered implicit foreground predictions and assigned soft labels. Setting $\delta_{lower}$ too high may result in only a few implicit foreground predictions available as soft labels, while setting it too low may lead to many false positives, mistaking background as implicit foreground. From Table 4, we chose $\delta_{lower} = 0.7$, which helps the model efficiently utilize implicit foreground predictions, especially in extremely low-shot scenarios (i.e., shot=1). Additionally, we noticed that when the FSOD detector performs well, it is not sensitive to the threshold $\delta_{lower}$. We speculate that this is because implicit foreground predictions are correctly assigned higher confidence scores, while background is given lower confidence scores.

### 4.4.3 Class prototypes update.
As mentioned earlier, we utilize the similarity between the ROI features of implicit foreground predictions and each class prototype to generate soft labels, where well-defined class prototypes can produce more accurate soft labels for implicit foreground predictions. However, since we initialize class prototypes using N-way K-shot data. During the teacher-student learning phase, predictions for the features of objects in each class are changing. Therefore, static prototype features cannot accurately represent their respective classes. We propose dynamically updating class prototypes using high-confidence pseudo-labels, aiming to gradually converge the prototypes to their true class distributions during test-time learning. In Table 5, we compare the results of static class prototypes and dynamically updated class prototypes across multiple samplings, with the latter showing consistent improvements. We observe that the improvement brought by dynamically updating class prototypes becomes more pronounced as the number of samples decreases.

### 4.4.4 Alternative data augmentation.
We also validated the data augmentation used for both the student and teacher networks. Generally, in semi-supervised object detection, weak augmentation is applied to input images for the teacher network, while strong augmentation is used for the student network. For details on data augmentation, readers are advised to refer to [28]. However, in our case, we found that even when employing weak data augmentation for the student network, its performance improved. As shown in

Table 6, consistently using weak-weak data augmentation enhanced performance across all settings. This is because, during test-time learning, we can only fine-tune on the test data for one epoch. Additionally, in scenarios of data scarcity, strong data augmentation disrupts the original data distribution, impeding the model convergence.

### 4.4.5 Qualitative evaluation.
We visualize the detection results of 1-shot of PASCAL VOC in Fig. 4. Our method can significantly alleviate the problem of classification confusion between base classes and novel classes. In the first column, DeFRCN misclassifies a base class (horse) as a novel class (cow), and in the second column, DeFRCN misclassifies a novel class (motorcycle) as a base class (bicycle). Our method addresses this issue through test-time learning. In the third column, DeFRCN predicts multiple local regions of a bus (novel class) as the bus category. Although we doesn't design any loss specifically for regression, the improvement in classification performance also helps the model alleviate this issue. Additionally, our method also improves the performance on base classes. For example, in column 4 of Fig. 4, DeFRCN incorrectly identifies a newsstand as a bottle and misses dense cars, both of which have been corrected by our method.

## 5 CONCLUSION

This paper proposes a novel framework for few-shot object detection, namely Prototype-based Soft-labels and Test-Time Learning (PS-TTL). It aims to address the challenge of accurately capturing the real data distribution under the condition of scarce samples from novel classes. To this end, we propose a Test-Time Learning (TTL) module to discover novel instances of test data, effectively alleviating the problem of overfitting to the distribution of base class. Furthermore, we design a Prototype-based Soft-labels (PS) strategy to unleash the potential of low-quality pseudo-labels, thereby significantly mitigating the constraints posed by few-shot samples. Extensive experiments are conducted on VOC and COCO, and PS-TTL reaches state-of-the-art performance, validating its effectiveness.

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
