# OpenReview forum: "PS-TTL: Prototype-based Soft-labels and Test-Time Learning for Few-shot Object Detection"
_acmmm.org/ACMMM/2024/Conference — MM2024 Poster_

### Official Review · Reviewer_ZLAP · 2024-05-21

**Rating:** 2
**Confidence:** 3

**Summary:**

The paper  proposes a new famework named Prototype-based Soft-labels and Test-Time Learning(PS-TTL) for few-shot object detection. The authors design a Test-Time Learning(TTL) module to discover novel instances on test data, and develop a Prototype-based Soft-labels (PS) strategy to unleash the potential of low-quality pseudo-labels. The authors believe it can enhance the accuracy capturing the real data distribution under the condition of scarce samples from novel classes.

**Strengths:**

The work introduces a strong and a lower bound confidence threshold remaining high-quality pseudo labels and generating  soft labels for images that have relatively low-confidence pseudo labels. This approach could utilize more input test images and improve the performance in the few-shot learning.

**Limitations:**

1. The authors mention that the weakness of mining implicit novel instances from the training set is that it relies on the assumption that unlabeled novel instances are widely present in abundant base which may not hold true in real-world scenarios. This is not articulated very clearly.  Figure 1(b) does not clearly elucidate the weakness of the approach either. Please clarify it specifically.

2. The authors mention that this work is the first attempt to explore fitting novel class data distributions in a way that is more in line with real-world scenarios. To my knowledge, several published papers have similar design before.  《Uncertainty-aware Pseudo Label Refinery for Domain Adaptive Semantic Segmentation》denoises pseudo labels by estimating uncertainty to a bimodal distribution without any manually designed thresholds. 《Consistent-Teacher: Towards Reducing Inconsistent Pseudo-targets in Semi-supervised Object Detection》 develops a Consistent-Teacher approach instead of the Mean-Teacher and sets adaptive assignment assigns anchor and dynamic thresholds to efficiently utilize the data. Please clarify the advantages of your work compared to these methods.

**Suitability:**

2

---

### Official Review · Reviewer_b6RX · 2024-05-21

**Rating:** 4
**Confidence:** 2

**Summary:**

This paper proposes a new PS-TTL method for FSOD to alleviate the sample scarcity issue of novel classes by taking advantage of the test-time samples with a well-designed test-time learning scheme. In particular, a teacher network which is EMA updated with the student network is used to produce high-confidence hard pseudo class labels for test-time samples while for those detection results with mild confidence, soft pseudo class labels are generated based on the similarities between the region features and the maintained class-wise prototypes. Then both kinds of pseudo labels are used in test-time learning for improving the generalization performance of baseline detectors. The experiments are conducted on two public datasets and PS-TTL shows improved performances compared to the baseline methods.

**Strengths:**

1.	This paper is well organized while the proposed method is well presented and is easy to follow.
2.	It is an appropriate and preferable way to consider both the hard and soft pseudo labels for test-time learning.
3.	The experimental results on VOC dataset show clear improvements of PS-TTL compared to the baseline methods. The authors also conduct extensive ablation studies to show the effect of each component in PS-TTL.

**Limitations:**

1.	The authors claim that the proposed method could be applied to different fine-tuning based baseline detectors, however only one existing baseline detector DeFRCN is considered in the experiments. I think more baseline detectors should also be considered to illustrate the effectiveness of PS-TTL.
2.	On MS COCO dataset, although DeFRCN+PS-TTL performs worse than LVC, I wonder will LVC+PS-TTL achieve a better performance than LVC.
3.	In some scenarios, the EMA updated teacher network will achieve a better test performance than the online updated student network at the end of training. Have the authors used the teacher network instead of the student network for testing and I wonder will the teacher network perform better.
4.	In Section 4.2, does the test-time learning be conducted on 2 samples and then these 2 samples are tested immediately? Does the batch size in this procedure influence the test performance?
5.	I recommend the authors to add a computational time comparison between DeFRCN and DeFRCN+PS-TTL to show the efficiency of the proposed method.
6.	I think the title of this paper needs to be further improved, as prototype-based soft-labels is indeed a part of the proposed test-time learning framework.
7.	There are some small typos and grammatical mistakes. For example, in Section 3.1, $C_b$ and $C^b$ are inconsistent. At line 334 and 849, the sentences are ended improperly.

**Suitability:**

2

---

### Official Review · Reviewer_eJmo · 2024-05-21

**Rating:** 5
**Confidence:** 3

**Summary:**

This paper targets on Few-Shot Object Detection (FSOD). Concretely, it proposed a framework called Prototype-based Soft-labels and Test-Time Learning (PS-TTL). They propose Test-Time Learning (TTL) to alleviate overfitting to base classes. It also develop prototype-based pseudo labels to enhance the quality of pseudo labels. Extensive experiment results prove the effectiveness of the proposed method.

**Strengths:**

1. This paper is overall well-written and easy to read. The figures are clear, demonstrating the idea in a good manner.
2. The method section is clear, the algorithm depicts the proposed method well.
3. I think the experiment results are abundant, with detailed analysis.

**Limitations:**

1. I think overall this paper is an ok submission to ACMMM. My major concern is that test-time learning and prototype-based method is not new to me (eg. they are widely applied in few-shot learning or transfer learning). So I need to see more discussions about the novelty of this paper. More discussion about the differences between the proposed method and previous methods are also welcome.

**Suitability:**

3

---

### Official Review · Reviewer_hXEE · 2024-05-23

**Rating:** 3
**Confidence:** 3

**Summary:**

Few-Shot Object Detection is a challenging task as it need to capture the distribution of unseen classes with quite limited instances. This paper proposed to address this problem by designing a novel framework named prototype-based soft-labels and test-time learning. The test-time learning module use a teacher network for self-training. Prototype-based soft-labels strategy asses the similarities between pseudo-labels ans category prototypes to unleash the potential of low quality pseudo-labels. It achieves good performance on several benchmarks.

**Strengths:**

The experimental results are satisfactory, demonstrating a commendable outcome. Using prototype-based soft-label method is a interesting way to introduce more fine-tuning samples to improve the performance. Test-Time Learning (TTL) that employs a mean-teacher network for self-training is proposed to discover novel instances on test data and Prototype based Soft-labels (PS) strategy is proposed  to unleash the potential of low-quality pseudo-labels.

**Limitations:**

1) Motivation Clarity: The paper fails to clearly explain why adopting a test-time learning method can address the shortcomings of previous methods in accurately capturing data distribution. The motivation behind this approach is not adequately explained.

2) Limited Novelty: The approach lacks novelty. [A] was the first to use a teacher and student model in test-time adaptation, and many subsequent works have followed this method. Hence, this is not a new approach.

3)The paper mentions that prototypes were initialized with novel training data, but it does not clarify how base classes are handled.

4) Comparison with Recent SOTA Methods: The paper does not compare its results with several recent state-of-the-art (SOTA) methods such as [B] and others. I hope these SOTA results will be supplemented or explanation of why they are not being compared.

5) Presentation of Results: The presentation of results in Table 2 can be potentially misleading. Highlighting data that is neither the best nor the worst in terms of performance could cause confusion.

6) Effectiveness of Prototype-Based Soft-Label Method: The paper needs to provide stronger evidence to convince readers of the effectiveness of the prototype-based soft-label method.


References:

[A] Wang, Qin, et al. "Continual test-time domain adaptation." Proceedings of the IEEE/CVF Conference on Computer Vision and Pattern Recognition. 2022.

[B] Zhang, Xinyu, Yuting Wang, and Abdeslam Boularias. "Detect everything with few examples." arXiv preprint arXiv:2309.12969 (2023).

**Suitability:**

2

---

### Meta-Review · Area_Chair_3JC4 · 2024-07-05

**Recommendation:** Accept (Poster)
**Confidence:** 5

**Metareview:**

This paper proposes to a Test-Time Learning (TTL) framework for Few-Shot Object Detection (FSOD), leveraging test data to improve the performance of FSOD. The idea of using test data to improve FSOD is novel and makes sense. The author did a great rebuttal. Most of the reviewers are positive about this work after the rebuttal, despite one reviewer not providing feedback. Therefore, my recommendation is Accept (Poster), and hope the authors can incorporate the reviewers' comments into the final version.